# Repurposing of Miltefosine as an Adjuvant for Influenza Vaccine

**DOI:** 10.3390/vaccines8040754

**Published:** 2020-12-11

**Authors:** Lu Lu, Carol Ho-Yan Fong, Anna Jinxia Zhang, Wai-Lan Wu, Iris Can Li, Andrew Chak-Yiu Lee, Thrimendra Kaushika Dissanayake, Linlei Chen, Ivan Fan-Ngai Hung, Kwok-Hung Chan, Hin Chu, Kin-Hang Kok, Kwok-Yung Yuen, Kelvin Kai-Wang To

**Affiliations:** 1State Key Laboratory for Emerging Infectious Diseases, Department of Microbiology, Carol Yu Centre for Infection, Li Ka Shing Faculty of Medicine, The University of Hong Kong, Pokfulam, Hong Kong, China; u3003963@connect.hku.hk (L.L.); chyfong@hku.hk (C.H.-Y.F.); zhangajx@hku.hk (A.J.Z.); hazelwu@hku.hk (W.-L.W.); canlee@hku.hk (I.C.L.); cyalee@hku.hk (A.C.-Y.L.); tkdissa@connect.hku.hk (T.K.D.); u3006707@connect.hku.hk (L.C.); chankh2@hku.hk (K.-H.C.); hinchu@hku.hk (H.C.); khkok@hku.hk (K.-H.K.); kyyuen@hku.hk (K.-Y.Y.); 2Department of Medicine, Li Ka Shing Faculty of Medicine, The University of Hong Kong, Pokfulam, Hong Kong, China; ivanhung@hku.hk

**Keywords:** influenza vaccine, vaccine, adjuvant, T_FH_, miltefosine

## Abstract

We previously reported that topical imiquimod can improve the immunogenicity of the influenza vaccine. This study investigated another FDA-approved drug, miltefosine (MTF), as a vaccine adjuvant. Mice immunized with an influenza vaccine with or without MTF adjuvant were challenged by a lethal dose of influenza virus 3 or 7 days after vaccination. Survival, body weight, antibody response, histopathological changes, viral loads, cytokine levels, and T cell frequencies were compared. The MTF-adjuvanted vaccine (MTF-VAC) group had a significantly better survival rate than the vaccine-only (VAC) group, when administered 3 days (80% vs. 26.7%, *p* = 0.0063) or 7 days (96% vs. 65%, *p* = 0.0041) before influenza virus challenge. Lung damage was significantly ameliorated in the MTF-VAC group. Antibody response was significantly augmented in the MTF-VAC group against both homologous and heterologous influenza strains. There was a greater T follicular helper cell (T_FH_) response and an enhanced germinal center (GC) reaction in the MTF-VAC group. MTF-VAC also induced both T_H_1 and T_H_2 antigen-specific cytokine responses. MTF improved the efficacy of the influenza vaccine against homologous and heterologous viruses by improving the T_FH_ and antibody responses. Miltefosine may also be used for other vaccines, including the upcoming vaccines for COVID-19.

## 1. Introduction

Pandemic, seasonal, and avian influenza viruses are associated with significant morbidity and mortality [1,2]. Currently available antivirals have limited efficacy in the treatment of influenza virus infection, especially for severe cases [3]. Vaccines have played a pivotal role in the prevention of influenza and have been shown to reduce complications [4]. The influenza vaccine alone, or with the pneumococcal vaccine, has been shown to reduce the incidence of myocardial infarction and stroke [5].

The currently available influenza vaccine only has an estimated overall efficacy of about 60% [4]. A meta-analysis showed that the pooled vaccine efficacy against influenza virus A(H3N2) is only 33% and is even lower against antigenically drifted strains [4]. Patients at older ages or with immunocompromised conditions have poorer responses [6]. Another problem with the currently available vaccines is that sufficient antibodies for protection only develop 2–3 weeks after vaccination.

Several strategies have been used to enhance and accelerate the immune response after influenza virus vaccination. These strategies include increasing the dose of the vaccine [7], vaccinating via the intradermal route [8], and adding adjuvant [6]. The only adjuvanted vaccine recommended by the US Centers for Disease Control and Prevention (CDC) for the 2019/2020 influenza season is FLUAD (Seqirus), which contains the oil-in-water emulsion MF59 adjuvant [9]. This vaccine is recommended for older adults aged 65 years or above. Although a meta-analysis including 31 studies reported MF59 improved influenza vaccine effectiveness [10], there is another meta-analysis that found no additional benefit in children aged 2 to 5 years old [11].

We have previously shown that intraperitoneal imiquimod, a toll-like receptor 7 (TLR7) agonist, can improve vaccine efficacy for both A(H1N1) and the avian influenza virus A(H7N9) in a mouse model [12,13]. The success of imiquimod in the mouse model has led to two randomized controlled trials, which have shown that topical application of imiquimod prior to intradermal influenza vaccination can augment, accelerate, and broaden the protective immune response [14,15].

Based on the success of imiquimod as a vaccine adjuvant, we sought to assess other FDA-approved drugs that can be used as an adjuvant. One such candidate is miltefosine (MTF), an FDA-approved anti-leishmaniasis drug. Studies have shown that MTF has several immunomodulatory functions. MTF stimulates TLR4 and TLR9 [16] but dampens NACHT, LRR and PYD domains-containing protein 3 (NLPR3) inflammasome assembly [17]. In monocytes, MTF could elicit expression of interferon-γ (IFN-γ), granulocyte-macrophage colony-stimulating factor (GM-CSF), intercellular adhesion molecule 1 (ICAM-1), and major histocompatibility complex (MHC) class I [18,19]. The anti-leishamanial effect of MTF depends on the IFN-γ pathway. In this study, we evaluated MTF as a potential vaccine adjuvant, and delineated the mechanism of action.

## 2. Materials and Methods 

### 2.1. Virus Strain

Mouse-adapted influenza A/HK/415742/09(H1N1) virus (415742Md) was grown in the 10-day old specific-pathogen-free chicken embryos. The antigenically-drifted strain A/HK/18M439315/18(H1N1) was cultured in Madin Darby canine kidney (MDCK) cells. The phylogenetic tree including both virus strains and multiple reference strains can be found in Appendix A. Inactivated 415742Md virus was obtained by treating by 0.03% (vol/vol) formaldehyde at 4 °C for 6 days. The completion of inactivation was confirmed by plaque assay.

### 2.2. Drugs

Miltefosine (MTF) was purchased from Sigma-Aldrich (St. Louis, MO, USA). It was dissolved in endotoxin-free water, aliquoted and stored at −20 °C until use. For immunization, 1 mg, 0.2 mg, and 0.008 mg were given to mice in 100 µL volume. The dosage of MTF was selected based on previous publications [20,21,22,23,24,25,26].

### 2.3. Influenza Vaccines

To test the MTF adjuvant effects more comprehensively and to eliminate the potential bias caused by vaccine preparations and immunogenicity, two different vaccines were used in each vaccination-challenge model. After observed similar immunogenicity of each vaccine, VAXIGRIP^®^ Trivalent influenza vaccine (2015 South hemisphere season, Lot: M7080-2) was used for the rest of the studies. The content and usage of each influenza vaccine is listed in Appendix A. All are split-virion vaccines that were prepared from influenza viruses propagated in embryonated eggs, and the quantity was 15 µg of hemagglutinin (HA) for each virus strain.

### 2.4. Animal Infection Experiments

Animal experiments were approved by the Animal Ethics Committee (CULATR No. 4111-16). Female BALB/c mice aged 6 to 8 weeks old were obtained from the Laboratory Animal Unit of the University of Hong Kong. The mice were housed in specific pathogen-free facilities with 12 h light–dark cycles and standard pellet feed and water ad libitum. Mice were immunized via intraperitoneal injection of influenza vaccine (3 µg hemagglutinin) with or without MTF adjuvant. For the control group, phosphate buffered saline (PBS) was used for the intraperitoneal injection. For challenge experiments, 427 PFU of 415742Md virus (10 × 50% lethal dose (LD_50_)) was inoculated via the intranasal route under ketamine (75 mg/kg) and xylazine (10 mg/kg) anesthesia. The predefined humane endpoint was 30% body weight loss or the mice being very inactive, having difficulty moving around, and accessing food and water. Mice that reached the endpoint were euthanized. The body weight and survival of infected mice were monitored daily for 14 days after virus inoculation.

### 2.5. Pulmonary Examinations

For histopathology examination, the left lung lobe of the mouse was fixed and embedded in paraffin blocks. Tissue blocks were sectioned and stained with hematoxylin and eosin (H&E). The H&E slides were examined under a light microscope for histopathology changes in a blinded manner.

The right lung lobes of the mouse were frozen and homogenized for viral load and cytokine level determinations. The viral loads were determined with the homogenate samples through plaque assay, as previously described [13]. Briefly, right lung lobes were frozen and homogenized. The lung homogenates were 10-fold serially diluted and inoculated in 300 µL in MDCK cells for 1 h at 37 °C. The cells were washed once and replenished with solid culture medium. The solid culture medium was prepared by mixing milted 1% (wt/vol) ultrapure low melting point agarose (Invitrogen, Corp., Carlsbad, CA, USA) with 2xMEM (Gibco^®^, ThermoFisher Scientific, Inc., Waltham, MA, USA) in a 1:1 ratio, and supplemented with 2 µg/mL N-tosyl-L-phenylalanine chloromethyl ketone (TPCK)-trypsin (Sigma-Aldrich, Corp., St. Louis, MO, USA) and 100U penicillin–streptomycin (Gibco^®^, ThermoFisher Scientific, Inc.). The plates were inverted and incubated for 72 h at 37 °C. After fixing with 10% neutral buffered formalin overnight, the agarose was removed, and the cells were stained with crystal violet for plaque counting.

For cytokine level determinations, the homogenate samples were lysed and further ground by going through the QIAshredder spin column (QIAGEN, Inc., Hilden, Germany). RNA was extracted by the RNeasy Mini Kit (QIAGEN, Inc., Hilden, Germany). The RNA was reverse transcribed into cDNA with Takara PrimeScript™ RT Master Mix (Perfect Real Time) (Takara Bio, Inc., Kusatsu, Shiga Prefecture, Japan). Quantitative PCR (qPCR) was performed with SYBR^®^ Premix Ex Taq™ (Tli RNase H Plus) (Takara Bio, Inc., Kusatsu, Shiga Prefecture, Japan) and specific primers on Roche LC96 (Roche, Inc., Basel, Switzerland). The primers used in this project are listed in Appendix A.

### 2.6. Antibody Titers

The hemagglutination inhibition assay (HAI), micro-neutralization assay (MN), and enzyme-linked immunosorbent assay (ELISA) assays were performed as previously described [13,27].

For HAI, 25 µL serum was mixed and incubated with 25 µL of 4 hemagglutinating unit (HAU) 415742Md virus for 1 h at room temperature. After 1 h, 50 µL 0.5% TRBC was added for determining the hemagglutination reaction of the mixture. For MN, 17.5 µL serum was mixed and incubated with 17.5 µL of 200 TCID_50_ 415742Md virus for 1 h at room temperature. Then, the mixture was inoculated in MDCK cells for 1 h at 37 °C. The cells were washed and replenished with MEM medium containing 2 µg/mL TPCK-trypsin and incubated for 72 h at 37 °C. The highest dilution of each mouse serum completely protecting the cell sheet from cytopathic effects (CPE) in 50% of the wells was taken to be the neutralizing antibody titer [28].

For the IgG, IgM, IgG1 and IgG2a ELISA assays, inactivated 415742Md virus was coated to the Nunc MaxiSorp™ flat-bottom plate (Invitrogen, Corp., Carlsbad, CA, USA) at 4 °C overnight. Mice serum samples were diluted and incubated on the plate at 4 °C overnight. Horseradish peroxidase (HRP)-conjugated anti-mouse IgG, IgM, IgG1, or IgG2a antibodies were applied for detection with 3,3′,5,5′-Tetramethylbenzidine (TMB) substrate. The OD450 was measured with Plate Reader (PerkinElmer, Inc., Waltham, MA, USA).

### 2.7. Splenocyte Single Cell Suspension Preparation

Spleens were harvested from mice and mashed through 40-micron Nylon cell strainers (Falcon^®^, Corning, Inc., New York, NY, USA). Flow-through cells were washed with complete RPMI 1640 (Gibco^®^, ThermoFisher Scientific, Inc., Waltham, MA, USA) containing 10% FBS and 100 U/mL penicillin–streptomycin. Red blood cells were lysed with Ammonium-Chloride-Potassium (AKC) lysing buffer (Gibco^®^, ThermoFisher Scientific, Inc., Waltham, MA, USA). Splenocytes were counted with a hemocytometer under the microscope for cell concentration determination.

### 2.8. T_FH_ Cells Flow Cytometry

Splenocytes were kept on ice through the whole process of staining. The cells were stained with Zombie™ dye and antibodies conjugated with fluorophores or biotins. The staining lasted for 30 min on ice. After the primary antibody staining, whenever necessary, streptavidin-conjugated allophycocyanin (APC) was added to cells and stained for another 30 min on ice. Cells were washed after all staining procedures and analyzed on BD LSRFortessa™ cell analyzer (BD Biosciences, Inc., Franklin Lakes, NJ, USA). T_FH_ cells were identified as CD3ε^+^CD4^+^CXCR5^+^CD279^+^. The raw data collected were gated in FlowJo™ 10.6.2 (LLC, BD Biosciences, Inc.). The antibodies and streptavidin conjugate used in the project are listed in Appendix A. Isotype controls of single staining and fluorescence-minus-one (FMO) controls were performed for each individual experiment.

### 2.9. GC Examination

Spleen samples were processed into frozen sections. The frozen sections were stained with APC-conjugated anti-mouse GL7 and PE-conjugated anti-mouse IgD (BioLegend, Inc., San Diego, CA, USA). DAPI dye (ThermoFisher Scientific, Inc., Waltham, MA, USA) was used as counterstain. Slides were observed under the Olympus BX53 fluorescent microscope (Olympus, Corp., Tokyo, Japan). Photos were taken and exported by CellSens Dimension (Olympus, Corp., Tokyo, Japan). Counting of follicles was performed with photos in a blinded manner to avoid bias.

### 2.10. Antigen-Specific Cytokine Responses

Mouse IFN-γ ELISpot^BASIC^ (HRP) kit (Mabtech, Inc., Stockholm, Sweden) together with MultiScreen-IP filter plates (MilliporeSigma, Inc., Burlington, MA, USA) or pre-coat Mouse IL-4 ELISpot^PLUS^ (ALP) kit (Mabtech, Inc., Stockholm, Sweden) were used following the manufacturer’s instructions for detection of cytokines. Splenocytes were seeded in at the concentration of 2.5 × 10^5^ cells per well. The influenza vaccine was used for stimulation. Combination of 20 ng/mL phorbol myristate acetate (PMA) (Sigma-Aldrich, St. Louis, MO, USA) and 1 µg/mL ionomycin calcium salt (Invitrogen, Carlsbad, CA, USA) was used as the positive control and medium only as the negative control. Cells were cultured at 37 °C for 19.5 h with stimulators. The plates were examined with CTL ImmunoSpot reader (Cellular Technology, Ltd., Cleveland, OH, USA).

### 2.11. Statistical Analysis

Data in this project are displayed as means or geometric means with the standard error of mean (SEM). For survival comparison, log–rank (Mantel–Cox) test was used. For body weight comparisons, multiple *t*-tests were applied. For pulmonary viral load, the statistical comparisons were conducted on logarithmized viral load with multiple *t*-tests. Relative expression levels of cytokine genes in each mouse were normalized by β-actin expression level and fold changes were calculated by comparing to naïve mice. Statistical comparisons of fold changes were conducted by two-way ANOVA. All statistical comparisons of antibody titers were conducted by two-way ANOVA. For HAI and MN titers, the undetectable samples were taken as titers of 5 for calculation. The titers were logarithmized before comparison. Frequencies of T_FH_ cells, frequencies of GCs, and numbers of antigen-specific cytokine secreting cells were compared by two-way ANOVA. Asterisk markers for significance levels: * for *p* < 0.05, ** for *p* < 0.01, *** for *p* < 0.001, and **** for *p* < 0.0001. All comparisons were conducted in GraphPad Prism 6.0 (GraphPad Software, Inc., San Diego, CA, USA). 

## 3. Results

### 3.1. MTF Adjuvant Improved Vaccine Efficacy in 7 Days

To determine the adjuvant effect of MTF, mice were immunized with an influenza vaccine with or without MTF, and were challenged by 415742Md 7 days after vaccination (Figure 1A). The MTF-VAC group (with 0.2 mg MTF) had a 96% survival rate, which was significantly higher than that of the 0.008 mg-MTF-VAC group (69%), VAC group (65%), MTF group (0%), and PBS group (0%) (Figure 1B). The survival of mice was not improved by increasing the MTF dose to 1 mg. The body weight of the MTF-VAC group was significantly better than the VAC group at 4 to 7 days post-infection (dpi) (Figure 1C). Significantly reduced lung damage in the MTF-VAC group was observed compared to the VAC group at 6 dpi (Figure 1D). The MTF-VAC group had less inflammatory cell infiltration and alveoli structure disruption, and had a more intact bronchiole epithelial cell layer than in the VAC group. The primary structures in distal lung areas of the MTF-VAC group were preserved, while the whole lung showed consolidation and damage for the VAC group.

At 2 dpi, the MTF-VAC group had significantly lower viral load compared with the PBS group (*p* < 0.05) (Figure 1E). Both the MTF-VAC and VAC groups had significantly lower viral loads compared with the PBS group at 6 dpi (*p* < 0.001). However, there was no significant difference in the viral load between the MTF-VAC group and the VAC group. Although there were no major differences in the expression levels of IFN-γ among all four groups, both the MTF-VAC and VAC groups had significantly lower IL-6 expression levels compared with the PBS group on 2 dpi and 4 dpi (*p* < 0.05) (Figure 1F). 

### 3.2. MTF Adjuvant Augmented and Broadened the Antibody Response of Influenza Vaccine When Given 7 Days before Viral Challenge 

Mice were infected 7 days after immunization and antibody response was assessed at 2, 4 and 6 dpi (Figure 2A). The geometric mean titers (GMTs) of HAI and MN were significantly higher in the MTF-VAC group at 6 dpi when compared to the MTF group and PBS group (Figure 2B). The anti-H1N1 IgM level at 4 dpi and anti-H1N1 IgG1 level at 6 dpi were significantly higher in the MTF-VAC group compared to the VAC group (*p* < 0.05 and *p* < 0.0001) (Figure 2C). The MTF-VAC group induced a more IgG1-biased antibody response compared to the VAC group, as indicated by a significantly higher IgG1/2a ratio at 6 dpi (*p* < 0.01) (Figure 2D). We also analyzed the antibody response 21 days after two doses of vaccination, given at day 0 and day 14. The MTF-VAC group had a significantly higher MN titer than the VAC group (Appendix A). 

To assess whether MTF improves antibody responses against antigenically-drifted virus, serum samples collected from mice post-infection were examined against antigenically-drifted 2018 H1N1 viruses. For mouse serum specimens collected at 6 dpi, the MN titer against the 2018 H1N1 was significantly higher in the MTF-VAC group than that of the VAC group (*p* < 0.001). There was no decrease in the MN titer of the MTF-VAC group against the 2018 H1N1 compared to the 2009 H1N1 (Figure 2E). The discrepancies in the performances of HAI and MN titers, namely the similar HAI titers of MTF-VAC group and VAC group against 2009 H1N1 and 2018 H1N1, is interesting as many studies have revealed that the HAI titer is insufficient in predicting protection in many conditions, [29,30,31] and MN assay is more sensitive in detecting the humoral immunity and predicting the protection against influenza virus [32,33].

### 3.3. MTF Adjuvant Improved Vaccine Efficacy in 3 Days 

To assess whether the onset of protective effect can be shortened, mice were infected 3 days after immunization (Figure 3A). The MTF-VAC group (80%) had a significantly higher survival rate than the VAC group (26.7%), MTF group (0%), and PBS group (0%). The survival rate of mice receiving 0.2 mg of MTF was higher than those receiving 0.008 mg of MTF (53.3%), although not reaching statistical significance (*p* = 0.1607). (Figure 3B) However, there was no significant difference in body weight loss between any two groups (Figure 3C).

There were no observable differences in the lung histopathological changes between the MTF-VAC and VAC group post-infection (photos not shown). There were no significant differences in the pulmonary viral loads of the MTF-VAC group and VAC group. Both groups had significantly lower viral loads compared to the PBS group at 4 and 6 dpi (*p* < 0.05 or *p* < 0.001). Both the MTF-VAC and VAC groups had significantly lower levels of IFN-γ expression when compared to the PBS group at 6 dpi (*p* < 0.001 and *p* < 0.01). The IL-6 expression levels were also significantly lower in the MTF-VAC and VAC groups compared to the PBS group at 2 and 6 dpi (*p* < 0.05 or *p* < 0.01). (Figure 3D).

At 4 dpi, the HAI, MN, total IgG and IgG1 titers of the MTF-VAC group were all significantly higher than that of the VAC group (*p* < 0.05 or *p* < 0.0001) (Figure 3E). The IgG1-bias was observed in the MTF-VAC group. The IgG1/2a ratios were significantly higher in the MTF-VAC group than the VAC group at both 4 and 6 dpi (*p* < 0.01 and *p* < 0.001) (Figure 3F).

### 3.4. MTF Adjuvant Significantly Increased T_FH_ Cell Frequency and Activity in the Spleen

Previous studies have shown that the TLR9 agonist can enhance the germinal center response, including an increase in the number of T_FH_ and GC B cells, and acceleration in the production of IgG [34]. Hence, we examined the splenic T cells after vaccination and after virus challenge (Figure 4A). The MTF-VAC group had higher frequencies of T_FH_ cells than the VAC group 7 days after vaccination (before infection) and 2 dpi (9 days after vaccination) (Figure 4B,C). At 7 days after vaccination, the frequency of T_FH_ cells in the MTF-VAC groups reached 0.92% of the total CD4^+^ T cells (95% CI 0.55–1.3%), which was significantly higher than that of VAC group (0.54%; 95% CI 0.39–0.69%) (*p* < 0.05). There was no significant difference between the VAC group and the PBS group (0.38%, 95% CI 0.23–0.53%) (*p* = 0.6053). At 2 dpi, the MTF-VAC group (2.1%, 95% CI 1.8–2.4%) had a significantly higher proportion of T_FH_ cells than VAC group (0.92%, 95% CI 0.72–1.1%), MTF group (0.49%, 95% CI 0.36–0.62%), and PBS group (0.41%, 95% CI 0.26–0.55%) (*p* < 0.0001). 

Next we examined the size and frequencies of GCs as a functional evaluation of T_FH_ cell activities. The MTF-VAC group had more GCs formed in spleens post-infection (Figure 4D). At 2 dpi, 76% (95% CI 41–110%) of B cell follicles (marked by IgD^+^) in the MTF-VAC group demonstrated GC activities (marked by GL7^+^). In the VAC group, only 37% (95% CI 36–38%) of B cell follicles showed GC activities. The GC reaction was significantly enhanced in the MTF-VAC group compared to the VAC group (*p* < 0.05). In addition, the areas of GCs were larger in the MTF-VAC group than the VAC group (Figure 4E). This demonstrated the increased frequencies of T_FH_ cells in the MTF-VAC group promoted the formation and activity of GCs.

### 3.5. MTF Adjuvanted Influenza Vaccine Enhanced the Antigen-Specific T_H_1 and T_H_2 Cytokine Responses

To assess influenza-specific cellular immune response, we compared the antigen-specific cytokine response of splenocytes from MTF-VAC and VAC group. The frequency of antigen-specific IFN-γ-secreting and IL-4-secreting cells upon in vitro antigen recall was significantly higher in the MTF-VAC group than those of the VAC group (Figure 5A,B). Increased frequencies of antigen-specific cytokine secreting cells indicated MTF-VAC induced more robust T_H_1 and T_H_2 antigen-specific immune responses in mice.

## 4. Discussion

The effectiveness of the influenza vaccine is suboptimal. Vaccine adjuvant has been used in boosting the immune response. In this study, we have found that MTF, an FDA-approved anti-leishmaniasis drug, significantly improved the protective efficacy of influenza vaccines. After the lethal challenge of A(H1N1) virus, the MTF-VAC group had a significantly higher survival rate than the VAC group. The improved survival in the adjuvanted group occurred as soon as 3 days after vaccination, and was accompanied by an accelerated and augmented antibody response, and a more robust T_FH_ cell response. Both T_H_1 and T_H_2 responses were stimulated. Furthermore, we have demonstrated that the MTF-adjuvanted vaccine induced a better antibody response against antigenically-drifted influenza viruses within the same subtype.

Generally, influenza vaccine studies assess the protective efficacy after at least two doses and at least 4 weeks after vaccination [35,36,37,38]. However, given the delayed availability of vaccines, patients would benefit from an early effect of vaccination. We assessed the early protective effect of the imiquimod-adjuvanted influenza vaccine previously [13,14]. In this study, we demonstrated an improved survival rate even when the mice were vaccinated as little as 3 days before infection. This quick onset of protection made MTF-adjuvanted influenza vaccine more potent in prevention of epidemic spreading.

The improved antibody response in the MTF-VAC group was associated with a more robust T_FH_ response and a more potent germinal center reaction. This is in contrast with our previous study on the TLR7 agonist, imiquimod, in which there was no difference in the GC reaction between the imiquimod-adjuvanted group and non-adjuvanted group [27]. T_FH_ cell is the key T cell subtype that interacts with B cells in lymphoid follicles [39]. T_FH_ cells are derived from CD4^+^ T cells in the T cell zone after being primed by dendritic cells. These T_FH_ cells migrate to the B cell zone and interact with B cells, promoting B cell differentiation into plasma cells and memory B cells [40]. T_FH_ cells secrete IL-21 to induce B cell activation, proliferation and differentiation. T_FH_ cells also express the costimulatory molecule CD40L for B cell survival. These interactions form the basis of GC reaction which can be seen in our MTF-VAC group. Live-attenuated influenza virus vaccine induced elevated antibody response was shown to correlate with early activation of T_FH_ cells [38]. Hill et al. also demonstrated that an adjuvanted malaria vaccine can increase circulating T_FH_ cells in patients to achieve better protection [37]. Dysfunction of T_FH_ cells can impair the antibody production after influenza vaccine administration [41]. Sage et al. has shown that older mice have a higher ratio of T_FR_ to T_FH_ cells, and that T_FH_ cells have defective antigen specific responses [42]. It remains to be determined whether MTF can improve T_FH_ cell functions for aged mice.

Both T_H_1 and T_H_2 responses are vital in vaccine-induced protection. Different adjuvants have differential effects on the T_H_1 and T_H_2 response. AS04, CpG, and CAF01 have been shown to induce a more T_H_1-biased response [43,44], while alum salts induce a better T_H_2 response [43]. Combined T_H_1 and T_H_2 responses were also observed in some adjuvant studies and found to be beneficial, such as MF59, MPL-A, and G3 [35,36,43]. Our study showed that MTF can improve both T_H_1 and T_H_2 response. Since T_H_1 are critical for cellular immune response while T_H_2 are important for antibody production [45], the improved protective effectiveness of MTF can be related to both types of helper T cells.

MTF may stimulate antigen-presenting cells, which in turn promote the differentiation of CD4^+^ T cells, as it is reported to be a TLR4 and TLR9 agonist [16]. Previous studies have shown that TLR4 activation by an adjuvant increases the production of proinflammatory cytokines and activation and antigen-loading on monocytes and dendritic cells, which translates into the activation of antigen-specific T and B cells, leading to stronger cellular and humoral immune responses [46]. For TLR9, a recent hepatitis B vaccine containing CpG, HEPLISAV-B^®^, has been licensed. After TLR9 stimulation, both antigen-presenting cell and B cell activation are induced. This also triggers a proinflammatory cytokine response that in turn leads to more potent cytotoxic T cell activation, higher antibody titers, as well as better cross-protection [47].

We showed that the increased in IgG response is mainly related to IgG1. Mouse IgG1 has been shown to prevent complement activation by IgG2a, ameliorating complement-mediated inflammatory damage [48]. The IgG1 subtype has also been shown to be an independent protective factor for septic shock [49]. IgG1 has been associated with neutralizing antibody response, while IgG2a was associated with clearance of influenza virus [50]. Therefore, the lack of difference in the level of IgG2a between MTF-VAC group and VAC group may explain why there was no difference in pulmonary viral load between these two groups. It should be noted that many successful vaccines or antivirals improved survival benefit but without a fast reduction in viral load [51,52,53,54,55]. 

Our study showed that MTF-VAC group has much less lung damage and inflammatory cell infiltration when compared with VAC group after being challenged by influenza virus. The H&E examination of lung tissue demonstrated that there was less alveolar damage in MTF-VAC group mice compared to that of the VAC post-infection. Several studies demonstrated that the blunted inflammatory response can be associated with a better outcome after virus challenge, independent of changes in viral load. In our previous studies with imiquimod adjuvant, there was no difference in viral load after viral challenge despite survival benefit [12,13]. The imiquimod adjuvant was later shown to have a much better immunogenicity in clinical trials [14,15]. Several studies have shown that survival correlates with inflammation rather than viral load [51,54].

There are now intensive efforts for developing influenza vaccines that can induce more potent and broader immune responses. It has been proposed that vaccination with conserved epitopes of influenza virus can result in broad protection. Universal vaccination has now been developed and is undergoing clinical trials [56,57]. Therefore, our strategy is valuable before the availability of an effective and safe universal influenza vaccine.

Our current study used a mouse model with intraperitoneal injection of adjuvant and influenza vaccine. In humans, inactivated or subunit influenza vaccines are given either intramuscularly or intradermally. For human use, miltefosine can either be incorporated into existing influenza vaccine preparation and injected either intramuscularly or intradermally, or can be applied topically on the skin prior to intradermal injection. A recent study showed that liposomal formulation of miltefosine was successful as a topical treatment of leishmania [58]. The potential required dosage of adjuvant could be achieved by currently available miltefosine cream [14,58,59,60]. Therefore, the evaluations and observations of miltefosine adjuvant on mice are likely to be transferrable to human topical adjuvant applications.

There are several limitations in this study. First, in this study, we have demonstrated that an improved T_FH_ cell response can lead to an enhanced antibody response. However, other pathways not involving T_FH_ cells may also play a role. For example, MTF can directly stimulate the differentiation of B cells into plasma cells. Second, this study used BALB/c mice aged 6–8 weeks. Since the adjuvanted vaccine is most beneficial among patients associated with poorer vaccine effectiveness, such as older adults and those with immunocompromised conditions, it remains to be determined whether MTF is beneficial in these populations. The cross-protection of MTF-VAC should be further tested with a heterologous virus challenge in mice as well. Furthermore, MTF’s impact on antigen-presenting cells need to be elucidated in the future, especially with previous reports of miltefosine stimulating TLR4 and TLR9 [16], which are frequently found on antigen-presenting cells such as dendritic cells and macrophages. Third, we administered miltefosine and the influenza vaccine via the intraperitoneal route instead of the intramuscular or intradermal route that is used in humans. The intraperitoneal route is commonly used for mice studies on vaccine adjuvants because of the ease of administration [61]. Our previous studies on imiquimod demonstrated that the results from the intraperitoneal adjuvant/vaccine model can guide clinical studies [14].

## 5. Conclusions

Our study showed that the FDA-approved MTF is a potent vaccine adjuvant. Since the systemic and topical safety profile of this drug is already known, MTF as a vaccine adjuvant can be assessed in clinical trials much sooner than other non-approved adjuvants. Our study highlighted the important role of T_FH_ in improving vaccine-induced immunogenicity. The results of this study go beyond the influenza vaccine. Miltefosine can also be evaluated for other vaccines, including the upcoming COVID-19 vaccines.

## Figures and Tables

**Figure 1 vaccines-08-00754-f001:**
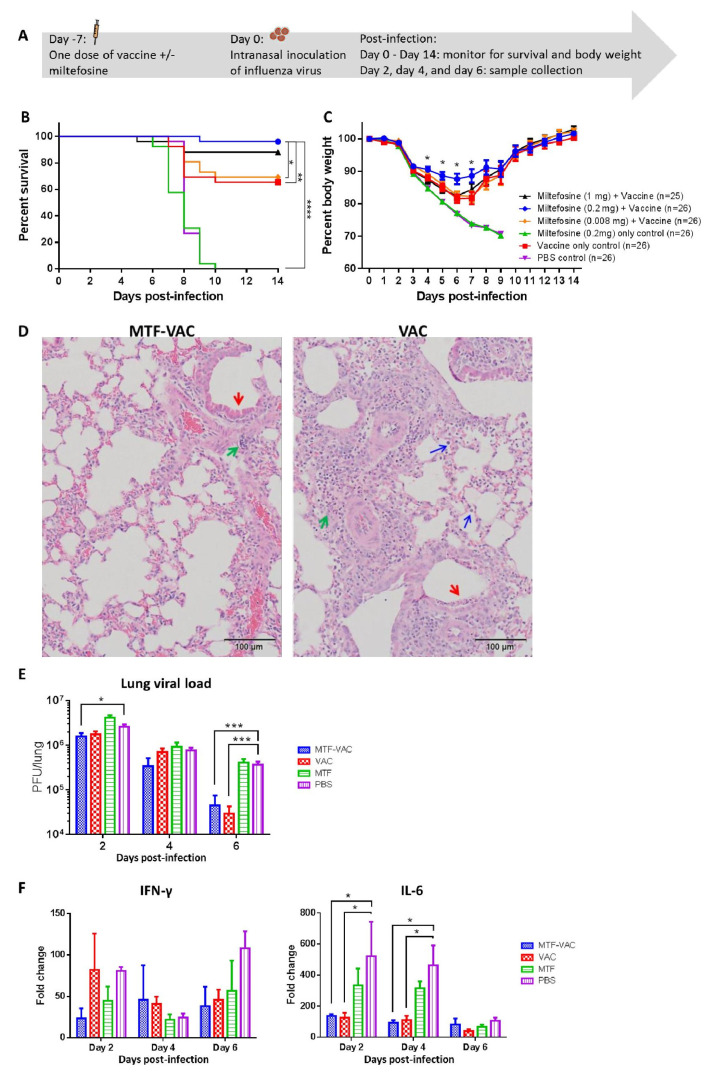
The effect of a single dose of vaccine on mice administrated 7 days before influenza virus challenge. Mice were vaccinated intraperitoneally with a split-virion influenza vaccine with or without miltefosine (MTF) in 200 µL. On the day of the virus challenge, 10LD_50_ of mouse-adapted A(H1N1)pdm09 virus was inoculated through the intranasal route in 40 µL. (**A**) Schedule of influenza vaccination, virus challenge, and sample collection. (**B**) Survival curve of mice. (**C**) Body weight curve of mice. (**D**) Representative hematoxylin and eosin (H&E) staining of lungs from MTF-adjuvanted vaccine (MTF-VAC) and VAC mice on 6 dpi; red arrows indicate epithelial cell layer damage in bronchioli; green arrows indicate lymphoid cell infiltration; blue arrows indicate monocytes and neutrophils entering alveolar space. (**E**) Pulmonary viral load post-infection (*n* = 6 per group). (**F**) Fold change of cytokine mRNA levels in lung tissue post-infection (*n* = 3 per group). For (**D**–**F**), MTF-VAC, miltefosine (0.2 mg) + vaccine group; VAC, vaccine only group; MTF, miltefosine (0.2 mg) only group; PBS, PBS only group. Data collected from 25–26 mice (4 independent experiments) for survival experiments. Log–rank (Mantel–Cox) test was used for survival comparison; multiple *t*-tests were used for body weight comparisons. Pulmonary viral load was determined by plaque assay and the statistical comparisons were conducted through multiple *t*-tests. Cytokine levels were assessed by RT-qPCR. Relative expression levels of target genes were normalized by β-actin expression level and fold change was calculated by comparing to naïve mice. Statistical comparisons were conducted by two-way ANOVA followed by a Tukey’s multiple comparison test. * for *p* < 0.05; *** for *p* < 0.001. Error bars represent standard error of the mean (SEM).

**Figure 2 vaccines-08-00754-f002:**
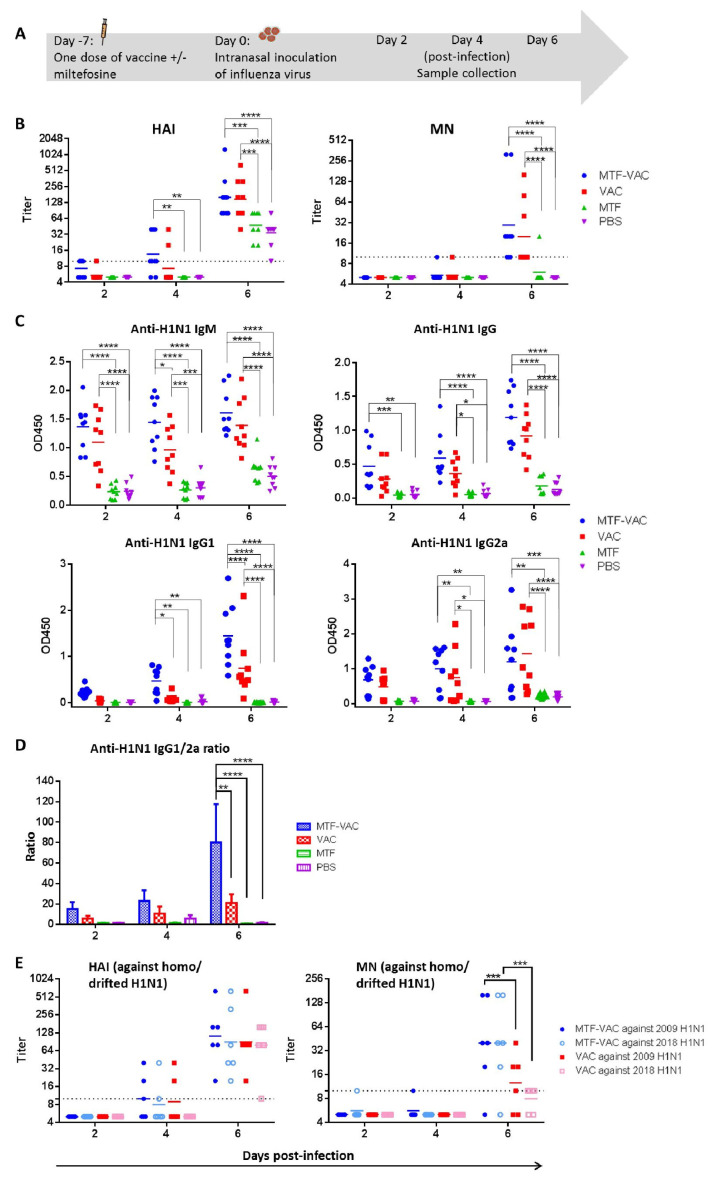
The effect of a single dose of vaccine with or without miltefosine on the antibody titers administrated 7 days before influenza virus challenge. (**A**) Schedule of influenza vaccination, virus challenge, and sample collection. (**B**) Hemagglutination inhibition assay (HAI) titers and micro-neutralization assay (MN) titers of serum samples post-infection. (**C**) Anti-H1N1 IgM, anti-H1N1 IgG, anti-H1N1 IgG1, and anti-H1N1 IgG2a antibody levels determined by ELISA. (**D**) Anti-H1N1 IgG1/G2a ratio. (**E**) HAI titers and MN titers of serum samples taken post-infection, assessed in vitro against homologous and antigenically-drifted H1N1 strains. MTF-VAC, miltefosine (0.2 mg) + vaccine group; VAC, vaccine only group; MTF, miltefosine (0.2 mg) only group; PBS, PBS only group. Short solid lines indicate geometric means of titers, and long dashed lines indicate the detection limit of HAI and MN assays. Data collected from 9 mice per group (3 independent experiments) for (**B**–**D**), and 6 mice per group (2 independent experiments) for (**E**). * for *p* < 0.05; ** for *p* < 0.01; *** for *p* < 0.001; **** for *p* < 0.0001, calculated by two-way ANOVA followed by a Tukey’s multiple comparison test.

**Figure 3 vaccines-08-00754-f003:**
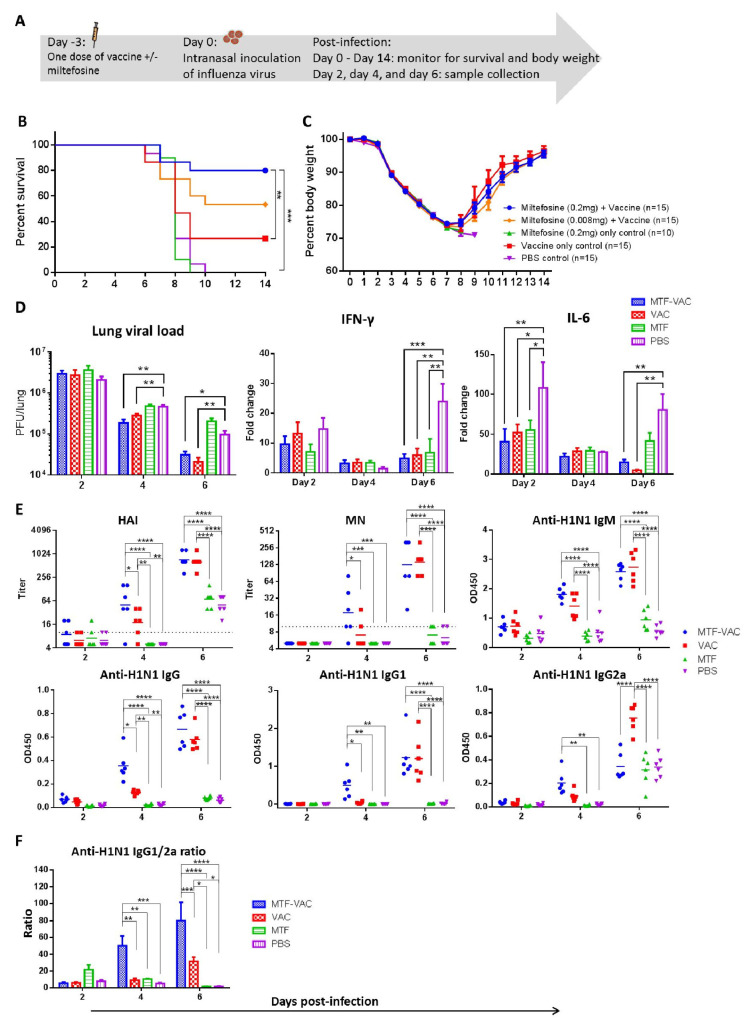
The effect of a single dose of vaccine on mice administrated 3 days before influenza virus challenge. Mice were vaccinated intraperitoneally with a split-virion influenza vaccine with or without miltefosine in 200 µL. On the day of the virus challenge, 10LD_50_ of mouse-adapted A(H1N1)pdm09 virus was inoculated through the intranasal route in 40 µL. (**A**) Schedule of influenza vaccination, virus challenge, and sample collection. (**B**) Survival curve of mice. (**C**) Body weight curve of mice. (**D**) Pulmonary viral load (*n* = 6 per group) and fold change of cytokine mRNA levels in lung tissue post-infection (*n* = 3 per group). (**E**) HAI titers and MN titers of serum samples post-infection; anti-H1N1 IgM, anti-H1N1 IgG, anti-H1N1 IgG1, and anti-H1N1 IgG2a antibody levels determined by ELISA (*n* = 6 per group). (**F**) Anti-H1N1 IgG1/G2a ratio (*n* = 6 per group). MTF-VAC, miltefosine (0.2 mg) + vaccine group; VAC, vaccine only group; MTF, miltefosine (0.2 mg) only group; PBS, PBS only group. Data collected from 10–15 mice (3 independent experiments) for survival experiments. Log–rank (Mantel–Cox) test was used for survival comparison; multiple *t*-tests were used for body weight comparisons. Pulmonary viral load was determined by plaque assay and compared through multiple *t*-tests. Cytokine levels were assessed by RT-qPCR. Relative expression levels of target genes were normalized by β-actin expression level and fold change was calculated by comparing to naïve mice. Statistical comparisons were conducted by two-way ANOVA followed by a Tukey’s multiple comparison test. Short solid lines indicate geometric means of titers, and long dashed lines indicate the detection limit of HAI and MN assays. All antibody titer comparisons were conducted by two-way ANOVA followed by a Tukey’s multiple comparison test. * for *p* < 0.05; ** for *p* < 0.01; *** for *p* < 0.001; **** for *p* < 0.0001. Error bars represent standard error of the mean (SEM).

**Figure 4 vaccines-08-00754-f004:**
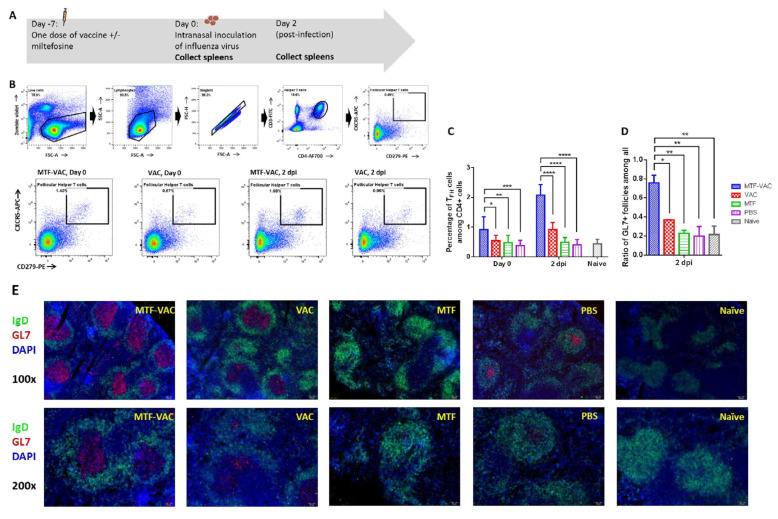
The effect of MTF-VAC and VAC on mice spleens before and after influenza virus challenge. Groups of mice were immunized 7 days before a lethal virus challenge with MTF-VAC, Vac, MTF, or PBS. At day 0, 10LD_50_ of the 415742Md virus was inoculated through the intranasal route. Spleen samples were collected on day 0 and 2 dpi. Flow cytometry was applied for assessing frequencies of T_FH_ cells with staining of Zombie Violet live/dead dye, anti-mouse CD3ε, anti-mouse CD4, anti-mouse PD-1, and anti-mouse CXCR5 antibodies. Frozen sections were prepared from 2 dpi spleen samples and stained with anti-mouse IgD and anti-mouse GL7 antibodies. The stained tissue slides were examined under the fluorescent microscope Olympus BX53. B cell follicles expressing IgD^+^GL7^−^ or IgD^+^GL7^+^ were counted with randomly renamed photos to avoid bias. (**A**) Schedule of influenza vaccine, virus challenge, and spleen sample collection. (**B**) Gating strategy and representative flow cytometry profiles of splenic T_FH_ cells obtained from mice. (**C**) T_FH_ cell frequencies in mice spleens assessed by flow cytometry. (**D**) Percentage of GL7^+^ B cell follicles (GC) among all B cell follicles (IgD^+^) in spleen. (**E**) Representative immunofluorescence staining of mouse spleen. MTF-VAC, miltefosine (0.2 mg) + vaccine group; Vac, vaccine only group; MTF, miltefosine (0.2 mg) only group; PBS, PBS only group. Data collected from 8 mice per group (3 independent experiments) for (**B**,**C**). Data were collected from 3 mice per group (1 independent experiment) for (**D**,**E**). * for *p* < 0.05; ** for *p* < 0.01; *** for *p* < 0.001, **** for *p* < 0.0001; calculated by two-way ANOVA followed by a Tukey’s multiple comparison test. Error bars represent standard error of the mean (SEM).

**Figure 5 vaccines-08-00754-f005:**
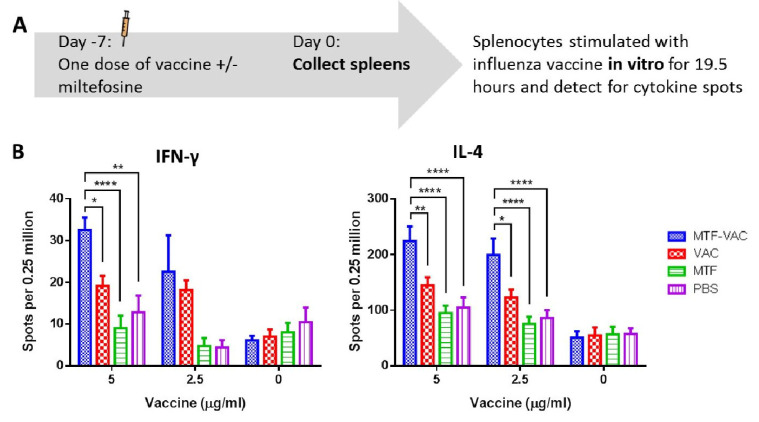
The antigen-specific cytokine responses of the splenocytes upon in vitro antigen recall. Spleens were harvested from mice 7 days after received a single dose of vaccine with or without miltefosine, and splenocytes were stimulated in vitro with the influenza vaccine for 19.5 h. (**A**) Schedule of influenza vaccine and spleen sample collection. (**B**) Number of IFN-γ secreting cells (left) and IL-4 secreting cells (right). MTF-VAC, miltefosine (0.2 mg) + vaccine group; VAC, vaccine only group; MTF, miltefosine (0.2 mg) only group; PBS, PBS only group. Data collected from 5–6 mice per group (2 independent experiments). * for *p* < 0.05; ** for *p* < 0.01; **** for *p* < 0.0001; calculated by two-way ANOVA followed by a Tukey’s multiple comparison test. Error bar indicates standard error of mean (SEM).

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
