# Peer review of "Repurposing of Miltefosine as an Adjuvant for Influenza Vaccine"

_vaccines, 2020, doi:10.3390/vaccines8040754_

Round 1

Reviewer 1 Report

This is a nicely written manuscript with FDA-approved MTF. I have two comments to improve the manuscript.

Currently influenza vaccine is given via intramuscularly, intradermally or intranasally in human. Please explain why intraperitoneal route was chosen in this rodent model and discuss potential limitations with route of injection of vaccine in human setting.

It is not clear what is the proposed next step to introduce this approach in human clinical setting. Is this just an idea or you are serious about using combination with human vaccine as both are approved in human use? Is the dose achievable in human? Please expand. If there is real human application, the importance of this manuscript is significant.

Author Response

Q: Currently influenza vaccine is given via intramuscularly, intradermally or intranasally in human. Please explain why intraperitoneal route was chosen in this rodent model and discuss potential limitations with route of injection of vaccine in human setting.

REPLY: We would like to thank the reviewer for the comments. Intraperitoneal injection of drugs or vaccine adjuvants is a common practice for animal experiments because of the ease of injection and the reliable dosage (Eisenbarth, Colegio, O’Connor, Sutterwala, & Flavell, 2008; Neirynck et al., 1999; Silva et al., 2017). Although intraperitoneal route is not used in humans, our previous studies on imiquimod demonstrated that the results from intraperitoneal adjuvant/vaccine model can guide clinical studies.

            For humans, the most common routes of vaccinations are intramuscular and subcutaneous routes. Polio vaccine can be given orally, while live attenuated influenza virus vaccine can be given intranasally. We have promoted the use of intradermal vaccination because of the presence of dermal dendritic cells which dramatically improves the immune response (Hung et al., 2016). Since intradermal vaccine can induce the best immune response, we chose this route, which can be further enhanced by topical adjuvants.

            We have added in lines 455-463: “Our current study used a mouse model with intraperitoneal injection of adjuvants and influenza vaccine. In humans, inactivated or subunit influenza vaccines are given either intramuscularly or intradermally. For human use, miltefosine can either be incorporated into existing influenza vaccine preparation and injected either intramuscularly or intradermally, or can be applied topically on the skin prior to intradermal injection. A recent study showed that liposomal formulation of miltefosine was successful as a topical treatment of Leishmania (Kavian et al., 2019). And the potential required dosage of adjuvant could be achieved by currently available miltefosine cream (Hung et al., 2016; Kavian et al., 2019; Terwogt, Mandjes, Sindermann, Beijnen, & ten Bokkel Huinink, 1999; de Bastiani et al., 2019). Therefore, the evaluations and observations of miltefosine adjuvant on mice are likely to be transferrable to human topical adjuvant applications.”

            And in lines 474-479: “Third, we administered miltefosine and influenza vaccine via the intraperitoneal route instead of the intramuscular or intradermal route that is used in humans. Intraperitoneal route is commonly used for mice studies on vaccine adjuvants because of the ease of administration (Eisenbarth, Colegio, O’Connor, Sutterwala, & Flavell, 2008). Our previous studies on imiquimod demonstrated that the results from intraperitoneal adjuvant/vaccine model can guide clinical studies (Hung et al., 2016).”

Eisenbarth, S. C., Colegio, O. R., O’Connor, W., Sutterwala, F. S., & Flavell, R. A. (2008). Crucial role for the Nalp3 inflammasome in the immunostimulatory properties of aluminium adjuvants. Nature, 453(7198), 1122-1126. doi:10.1038/nature06939

de Bastiani, F., Spadari, C. C., de Matos, J. K. R., Salata, G. C., Lopes, L. B., & Ishida, K. (2019). Nanocarriers Provide Sustained Antifungal Activity for Amphotericin B and Miltefosine in the Topical Treatment of Murine Vaginal Candidiasis. Front Microbiol, 10, 2976. doi:10.3389/fmicb.2019.02976

Hung, I. F., Zhang, A. J., To, K. K., Chan, J. F., Li, P., Wong, T. L., . . . Yuen, K. Y. (2016). Topical imiquimod before intradermal trivalent influenza vaccine for protection against heterologous non-vaccine and antigenically drifted viruses: a single-centre, double-blind, randomised, controlled phase 2b/3 trial. Lancet Infect Dis, 16(2), 209-218. doi:10.1016/S1473-3099(15)00354-0

Kavian, Z., Alavizadeh, S. H., Golmohamadzadeh, S., Badiee, A., Khamesipour, A., & Jaafari, M. R. (2019). Development of topical liposomes containing miltefosine for the treatment of Leishmania major infection in susceptible BALB/c mice. Acta Trop, 196, 142-149. doi:10.1016/j.actatropica.2019.05.018

Neirynck, S., Deroo, T., Saelens, X., Vanlandschoot, P., Jou, W. M., & Fiers, W. (1999). A universal influenza A vaccine based on the extracellular domain of the M2 protein. Nature Medicine, 5(10), 1157-1163. doi:10.1038/13484

Silva, M., Nguyen, T. H., Philbrook, P., Chu, M., Sears, O., Hatfield, S., . . . Sitkovsky, M. V. (2017). Targeted Elimination of Immunodominant B Cells Drives the Germinal Center Reaction toward Subdominant Epitopes. Cell Rep, 21(13), 3672-3680. doi:10.1016/j.celrep.2017.12.014

Q: It is not clear what is the proposed next step to introduce this approach in human clinical setting. Is this just an idea or you are serious about using combination with human vaccine as both are approved in human use? Is the dose achievable in human? Please expand. If there is real human application, the importance of this manuscript is significant.

REPLY: We would like to thank the reviewer for the questions. As with the experience from imiquimod, we plan to proceed to further studies on the use of miltefosine as a vaccine adjuvant for human studies. For human use, miltefosine can either be incorporated into existing influenza vaccine preparation and injected either intramuscularly or intradermally, or can be applied topically onto the skin prior to injection of the influenza vaccine. A clinical study used 6% miltefosine cream, which is equivalent to 6 mg of miltefosine per 100 ml of cream (Terwogt, Mandjes, Sindermann, Beijnen, & ten Bokkel Huinink, 1999). This dose is much higher than used in the dosage used for our intraperitoneal injection (0.2 mg). A recent study showed that liposomal formulation of miltefosine (up to 4%) was successful as a topical treatment of Leishmania (Kavian et al., 2019).

            We have added in lines 455-463: “Our current study used a mouse model with intraperitoneal injection of adjuvants and influenza vaccine. In humans, inactivated or subunit influenza vaccines are given either intramuscularly or intradermally. For human use, miltefosine can either be incorporated into existing influenza vaccine preparation and injected either intramuscularly or intradermally, or can be applied topically on the skin prior to intradermal injection. A recent study showed that liposomal formulation of miltefosine was successful as a topical treatment of Leishmania (Kavian et al., 2019). And the potential required dosage of adjuvant could be achieved by currently available miltefosine cream (Hung et al., 2016; Kavian et al., 2019; Terwogt, Mandjes, Sindermann, Beijnen, & ten Bokkel Huinink, 1999; de Bastiani et al., 2019). Therefore, the evaluations and observations of miltefosine adjuvant on mice are likely to be transferrable to human topical adjuvant applications.”

            And in lines 474-479: “Third, we administered miltefosine and influenza vaccine via the intraperitoneal route instead of the intramuscular or intradermal route that is used in humans. Intraperitoneal route is commonly used for mice studies on vaccine adjuvants because of the ease of administration (Eisenbarth, Colegio, O’Connor, Sutterwala, & Flavell, 2008). Our previous studies on imiquimod demonstrated that the results from intraperitoneal adjuvant/vaccine model can guide clinical studies (Hung et al., 2016).”

Eisenbarth, S. C., Colegio, O. R., O’Connor, W., Sutterwala, F. S., & Flavell, R. A. (2008). Crucial role for the Nalp3 inflammasome in the immunostimulatory properties of aluminium adjuvants. Nature, 453(7198), 1122-1126. doi:10.1038/nature06939

de Bastiani, F., Spadari, C. C., de Matos, J. K. R., Salata, G. C., Lopes, L. B., & Ishida, K. (2019). Nanocarriers Provide Sustained Antifungal Activity for Amphotericin B and Miltefosine in the Topical Treatment of Murine Vaginal Candidiasis. Front Microbiol, 10, 2976. doi:10.3389/fmicb.2019.02976

Hung, I. F., Zhang, A. J., To, K. K., Chan, J. F., Li, P., Wong, T. L., . . . Yuen, K. Y. (2016). Topical imiquimod before intradermal trivalent influenza vaccine for protection against heterologous non-vaccine and antigenically drifted viruses: a single-centre, double-blind, randomised, controlled phase 2b/3 trial. Lancet Infect Dis, 16(2), 209-218. doi:10.1016/S1473-3099(15)00354-0

Kavian, Z., Alavizadeh, S. H., Golmohamadzadeh, S., Badiee, A., Khamesipour, A., & Jaafari, M. R. (2019). Development of topical liposomes containing miltefosine for the treatment of Leishmania major infection in susceptible BALB/c mice. Acta Trop, 196, 142-149. doi:10.1016/j.actatropica.2019.05.018

Neirynck, S., Deroo, T., Saelens, X., Vanlandschoot, P., Jou, W. M., & Fiers, W. (1999). A universal influenza A vaccine based on the extracellular domain of the M2 protein. Nature Medicine, 5(10), 1157-1163. doi:10.1038/13484

Silva, M., Nguyen, T. H., Philbrook, P., Chu, M., Sears, O., Hatfield, S., . . . Sitkovsky, M. V. (2017). Targeted Elimination of Immunodominant B Cells Drives the Germinal Center Reaction toward Subdominant Epitopes. Cell Rep, 21(13), 3672-3680. doi:10.1016/j.celrep.2017.12.014

Reviewer 2 Report

Lu et al. explored the use of miltefosine (MTF) for immunologic adjuvant. The immunological analysis is robust, and the manuscript is well written.

The manuscript could be further improved if the authors clarify the following points.

  • The authors claim that “MTF improved the efficacy of the influenza vaccine against homologous and heterologous viruses”, which is found in the abstract section. HAI assay results show that humoral responses for drifted H1N1 are similar, while MN assay results show improved responses in the MTF-VAC group as compared to the VAC group. Since there is only a modest difference, cross protection should be examined not only neutralizing antibody, but also change study using heterologous virus to justify this conclusion.

  • Most of assay results show humoral responses between the MTF-VAC and VAC groups are similar except anti-H1N1 IgM level at 4 dpi and anti-H1N1 IgG1 level at 6 dpi, but this difference may explain the improvements reported in protection and lung pathology in this study. The authors analyzed antibody responses at 3 and 7 days after vaccination to determine if the MTF accelerates the responses. However, the adjuvant effect of MTF would not be fully characterized in this analysis only. The effects of adjuvant on the process of B cell differentiation and maturation requires additional 2-3 weeks. It would be informative if the authors present immune correlate data at 21 or 28 days after vaccination, if they have.

  • Figure captions for Figure 1 needs to be improved. In Figure 1A-1C, multiple MTF doses were used for MTF-VAC groups. However, there is a following statement “MTF-VAC, miltefosine (0.2 mg) + vaccine group; Vac, vaccine only group; MTF, miltefosine (0.2 mg) only group; PBS, PBS only group”. It is unclear which study used this grouping. It appears that this grouping is for Figure 1D and 1E only, where the lung damage study (Figure 1D) and lung viral load study (Figure 1E) were performed. Please clarify.

  • The mechanism(s) of actin for MTF’s anti-leishmaniasis effect is not clear. As such, its molecular mechanisms of action for the adjuvant effect are not clearly defined in this work. The potential mechanisms of MTF should be discussed more in detail in the discussion section. In particular, please discuss on MTF’s significant TH2 bias. In the introduction section, the authors note that “MTF stimulates TLR4 and TLR9”. This statement could be further extended to the potential mechanisms for the bias with MTF.

Author Response

Q: The authors claim that “MTF improved the efficacy of the influenza vaccine against homologous and heterologous viruses”, which is found in the abstract section. HAI assay results show that humoral responses for drifted H1N1 are similar, while MN assay results show improved responses in the MTF-VAC group as compared to the VAC group. Since there is only a modest difference, cross protection should be examined not only neutralizing antibody, but also change study using heterologous virus to justify this conclusion.

REPLY: Thank you for the suggestion. We agree with the reviewer that the best way to illustrate cross-protection is to challenge mice with a heterologous virus. We have added this into the limitation section as lines 470-471: “The cross-protection of MTF-VAC should be further tested with a heterologous virus challenge in mice as well.”

Q: Most of assay results show humoral responses between the MTF-VAC and VAC groups are similar except anti-H1N1 IgM level at 4 dpi and anti-H1N1 IgG1 level at 6 dpi, but this difference may explain the improvements reported in protection and lung pathology in this study. The authors analyzed antibody responses at 3 and 7 days after vaccination to determine if the MTF accelerates the responses. However, the adjuvant effect of MTF would not be fully characterized in this analysis only. The effects of adjuvant on the process of B cell differentiation and maturation requires additional 2-3 weeks. It would be informative if the authors present immune correlate data at 21 or 28 days after vaccination, if they have.

REPLY: We would like to thank the reviewer for the insightful suggestions. We have added the data of the serological studies conducted on day 21 after vaccination (see new supplementary figure S2). Miltefosine improved the immunogenicity of influenza vaccine. HAI and MN assays showed only mice in the MTF-VAC group had detectable antibody levels against 415742Md on day 14 and day 21 after vaccination. MTF-VAC group had a significantly high MN titer than the VAC group. This has been added in the section 3.2 (lines 251-253) as follow: “We have also analyzed the antibody response 21 days after 2 doses of vaccination, given at day 0 and day 14. MTF-VAC group had a significantly high MN titer than the VAC group (Supplementary Figure S2).”

The figure was added in Supplementary materials on page 6 with figure description “Supplementary Figure S2 The effect of two doses of vaccines with or without miltefosine on the antibody titers without influenza virus challenge. (A) Schedule of influenza vaccination and sample collection. (B) HAI titers and MN titers of serum samples after vaccination. (C) Anti-H1N1 IgM and anti-H1N1 IgG antibody levels determined by ELISA.

MTF-VAC, miltefosine (0.2 mg) + vaccine group; Vac, vaccine only group; MTF, miltefosine (0.2 mg) only group; PBS, PBS only group. Short solid lines indicate geometric means of titers, and long dashed lines indicate the detection limit of HAI and MN assays. Data collected from 3 mice per group. * for P<0.05; **** for P<0.0001, calculated by two-way ANOVA followed by a Tukey’s multiple comparison test.”

Q: Figure captions for Figure 1 needs to be improved. In Figure 1A-1C, multiple MTF doses were used for MTF-VAC groups. However, there is a following statement “MTF-VAC, miltefosine (0.2 mg) + vaccine group; Vac, vaccine only group; MTF, miltefosine (0.2 mg) only group; PBS, PBS only group”. It is unclear which study used this grouping. It appears that this grouping is for Figure 1D and 1E only, where the lung damage study (Figure 1D) and lung viral load study (Figure 1E) were performed. Please clarify.

REPLY: We would like to thank the reviewer for the comment, and we have now updated the figure legend as follow “For (D) to (F), MTF-VAC, miltefosine (0.2 mg) + vaccine group; Vac, vaccine only group; MTF, miltefosine (0.2 mg) only group; PBS, PBS only group”.

Q: The mechanism(s) of actin for MTF’s anti-leishmaniasis effect is not clear. As such, its molecular mechanisms of action for the adjuvant effect are not clearly defined in this work. The potential mechanisms of MTF should be discussed more in detail in the discussion section. In particular, please discuss on MTF’s significant TH2 bias. In the introduction section, the authors note that “MTF stimulates TLR4 and TLR9”. This statement could be further extended to the potential mechanisms for the bias with MTF.

REPLY: We would like to thank the reviewer for the suggestion on further elaboration of the mechanism of miltefosine.

  1. In this study, we showed that MTF induce both TH1 and TH2 response, as evidenced by the increase in IFN-γ and IL-4, respectively.
  2. For TLR4, previous study showed that TLR4 activation by adjuvant increases the production of proinflammatory cytokines and activation and antigen-loading on monocytes and dendritic cells, which translates into the activation of antigen-specific T and B cells, leading to stronger cellular and humoral immune responses (Didierlaurent et al., 2009).
  3. For TLR9, a recent hepatitis B vaccine containing CpG, Heplisav-B®, has been licensed. After TLR9 stimulation, both antigen-presenting cell and B cell activation are induced. This also triggers a proinflammatory cytokine response that in turn leads to more potent cytotoxic T cell activation, higher antibody titers, as well as better cross-protection (Klinman, 2004).

            We have now added the following in the discussion section lines 423-431: “MTF may stimulate antigen-presenting cells, which in turn promote the differentiation of CD4+ T cells, as it is reported to be a TLR4 and TLR9 agonist (Mukherjee et al., 2012). Previous study showed that TLR4 activation by adjuvant increases the production of proinflammatory cytokines and activation and antigen-loading on monocytes and dendritic cells, which translates into the activation of antigen-specific T and B cells, leading to stronger cellular and humoral immune responses (Didierlaurent et al., 2009). For TLR9, a recent hepatitis B vaccine containing CpG, Heplisav-B®, has been licensed. After TLR9 stimulation, both antigen-presenting cell and B cell activation are induced. This also triggers a proinflammatory cytokine response that in turn leads to more potent cytotoxic T cell activation, higher antibody titers, as well as better cross-protection (Klinman, 2004).”

            And amended the sentence in lines 471-474 to: “Furthermore, MTF’s impacts on antigen-presenting cells are needed to be elucidated in the future, especially with previous report of miltefosine stimulating TLR4 and TLR9 (Mukherjee et al., 2012), which are frequently found on antigen-presenting cells such as dendritic cells and macrophages.”

            The sentence in lines 400-402 “Being a TLR9 agonist, MTF can stimulate dendritic cells, which in turn promote the differentiation of CD4+ T cells into TFH cells.” was removed.

Didierlaurent, A. M., Morel, S., Lockman, L., Giannini, S. L., Bisteau, M., Carlsen, H., . . . Garcon, N. (2009). AS04, an aluminum salt- and TLR4 agonist-based adjuvant system, induces a transient localized innate immune response leading to enhanced adaptive immunity. J Immunol, 183(10), 6186-6197. doi:10.4049/jimmunol.0901474

Klinman, D. M. (2004). Immunotherapeutic uses of CpG oligodeoxynucleotides. Nature Reviews Immunology, 4(4), 249-259. doi:10.1038/nri1329

Mukherjee, A. K., Gupta, G., Adhikari, A., Majumder, S., Kar Mahapatra, S., Bhattacharyya Majumdar, S., & Majumdar, S. (2012). Miltefosine triggers a strong proinflammatory cytokine response during visceral leishmaniasis: role of TLR4 and TLR9. Int Immunopharmacol, 12(4), 565-572. doi:10.1016/j.intimp.2012.02.002

Reviewer 3 Report

In this manuscript, Lu et al evaluate the benefits of using miltefosine as an adjuvant for influenza vaccine. Compared to vaccination without this adjuvant, the results presented demonstrate a better efficiency of MTF-VAC over VAC. The effects are even visible at day 3 post infection which is an improvement over vaccine without this adjuvant.  The results are clearly presented, though not very original since this study follows a similar one by the authors with imiquimod as added adjuvant. However they show an important difference with miltefosine enhancing the response by T follicular helper cells.

Minor point:

The discussion is clear and point out the limitations of this study. In particular, this study was conducted on young animals and it is known that aged TFH cells have functional defects. This feature would severely limits the use of this adjuvant. Though it is discussed, a reference about TFH and aging should be added, as PUBMED ID 26146074 by Sage et al 2015, or any appropriate reference.

Author Response

Q: The discussion is clear and point out the limitations of this study. In particular, this study was conducted on young animals and it is known that aged TFH cells have functional defects. This feature would severely limits the use of this adjuvant. Though it is discussed, a reference about TFH and aging should be added, as PUBMED ID 26146074 by Sage et al 2015, or any appropriate reference.

REPLY: We thank the reviewer for highlighting the impact of aging on the TFH cell function. We have now added the following sentence in the discussion section: “It is also reported that changes in ratio and cell characteristics of aged TFH cells may be a main contributor to the waned humoral immunity in aged mice (Sage, Tan, Freeman, Haigis, & Sharpe, 2015). It remains to be determined whether MTF can improve TFH cell functions for aged mice”.

Sage, P. T., Tan, C. L., Freeman, G. J., Haigis, M., & Sharpe, A. H. (2015). Defective TFH Cell Function and Increased TFR Cells Contribute to Defective Antibody Production in Aging. Cell Rep, 12(2), 163-171. doi:10.1016/j.celrep.2015.06.015

Round 2

Reviewer 2 Report

The authors addressed all the concerns raised by the reviewer. The manuscript is acceptable.